# Isolation and Characterization of *Bacillus* Strains from Egyptian Mangroves: Exploring Their Endophytic Potential in Maize for Biological Control of *Spodoptera frugiperda*

**DOI:** 10.3390/biology13121057

**Published:** 2024-12-17

**Authors:** Hayam M. Fathy, Mona Awad, Nawal A. Alfuhaid, El-Desoky S. Ibrahim, Moataz A. M. Moustafa, Ayatollah S. El-Zayat

**Affiliations:** 1Department of Microbiology, Faculty of Agriculture, Cairo University, Giza 12613, Egypt; hayam.fathy@cu.edu.eg (H.M.F.); ayatollah.elzayat@agr.cu.edu.eg (A.S.E.-Z.); 2Department of Economic Entomology and Pesticides, Faculty of Agriculture, Cairo University, Giza 12613, Egypt; aldosokyibrahim@yahoo.com (E.-D.S.I.); moat_mon@agr.cu.edu.eg (M.A.M.M.); 3Department of Biology, College of Science and Humanities, Prince Sattam bin Abdulaziz University, Al-Kharj 11942, Saudi Arabia; n.alfuhaid@psau.edu.sa

**Keywords:** colonization, endophyte bacteria, insecticidal activity, marine *Bacillus* strains, the fall armyworm

## Abstract

This study addresses the growing problems of pesticide usage, such as environmental pollution and pest resistance. The Fall Armyworm (*Spodoptera frugiperda*) is an insect known for damaging maize crops and withstanding chemical treatments. Traditional pesticides have become less effective, leading researchers to explore natural, bacteria-based solutions. The study aimed to test if certain types of bacteria found in marine environments could live within maize plants and protect them from this pest. In our study, these bacteria were applied to the plants in four different ways: spraying on leaves, treating seeds, drenching soil, and a combination of all three. The results showed that the bacteria, particularly *Bacillus* sp. Esh39 and *Bacillus tequilensis* R39, caused significant mortality in laboratory tests. However, while effective in reducing pests, the bacteria did not notably boost plant growth. The study highlights that using these natural bacteria could be a sustainable alternative to chemical pesticides, offering new ways to protect crops and lessen environmental impact. Further research could strengthen this natural method of insect control, potentially benefiting agriculture and reducing dependency on synthetic pesticides.

## 1. Introduction

The Fall Armyworm (FAW), scientifically known as *Spodoptera frugiperda* (JE Smith) (family Noctuidae, order Lepidoptera), is a widespread crop pest species that inflicts significant damage to maize crops [1,2]. FAW has historically been a problem in the Americas but has recently emerged as a worldwide concern for food production [3]. In Africa, the initial documentation occurred in Nigeria in 2016 [2]. Currently, it has been identified in over 44 additional African nations and many Asian countries, including Nepal, India, and China [4,5,6,7]. The FAW poses a substantial risk to food security and the sustainability of crop yields due to ineffective management practices, its swift proliferation, and its potential to inflict considerable harm on agricultural products [8]. The FAW can reduce maize production by up to 70% when it attacks corn plants during their early growth stages [9]. Hence, [10] documented losses in maize production about 40–70% in Kenya and Ethiopia. The European Union issued Commission Decision (EU) 2018/638 on emergency measures to prevent the introduction and spread of FAW within the EU [11].

Typically, control of FAW is achieved through the application of synthetic insecticides. However, excessive use of these pesticides has resulted in resistance to several insecticide classes, including pyrethroids, organophosphates, and carbamates [12,13,14]. This resistance necessitates higher volumes of insecticides than those required for other pest species [15,16] and often demands multiple applications to manage existing FAW populations [17]. Such reliance on chemical control has significant environmental impacts, reduces populations of beneficial insects, and poses threats to food safety. Furthermore, the efficacy of these insecticides against mature FAW larvae is limited due to the pest’s cryptic feeding behavior [18]. Consequently, managing FAW is challenging, as it currently relies heavily on chemical insecticides. In addition, the indiscriminate use of various insecticides can adversely affect the ecosystem and its inhabitants [19]. Additionally, FAW has developed resistance to transgenic *Bt*-maize [20]. In this context, the efficacy of biopesticides has emerged as a sustainable alternative for controlling harmful pests. The advancement of biopesticides focuses on targeting specific pests while promoting agricultural production without compromising plant development [21].

Entomopathogens, naturally occurring regulators of insect pest populations, present another option for controlling FAW. The *Bacillus* genus exhibits substantial genetic diversity, with strains in various environments ranging from seawater to soil and even in extreme conditions [22]. *Bacillus* species are valuable in agriculture due to their ability to produce a range of lipopeptides—bacillomycin, surfactin, iturin, and fengycin—which exhibit activity against insects, mites, nematodes, and plant pathogens [23,24]. *Bacillus* spp. are Gram-positive, aerobic, spore-forming bacteria that produce parasporal crystals during sporulation, releasing protein toxins effective against various insect groups with high host specificity while ensuring environmental safety [25].

The *Bacillus* spp. usage has grown recently for protecting high-value crops, where it is essential to ensure a high degree of safety, selectivity, and effective resistance management [26]. However, the commercial utilization of *Bacillus* is constrained by its vulnerability to abiotic factors such as temperature, humidity, and ultraviolet radiation. To address these challenges, researchers have explored the ecological role of entomopathogenic bacteria as endophytes. As endophytes, these bacteria live symbiotically within plant tissues, gaining protection from abiotic stresses and UV exposure while also providing defense against insect pests [27]. Though endophyte-based pest management shows promise, research in this area is still in its early stages, with only a few studies demonstrating its potential as a novel pest control strategy [28].

Several factors contribute to the attractiveness of employing endophytes as biological control agents: (i) they offer a natural method of pest control, thereby decreasing dependence on harmful chemicals [29]; (ii) they are potentially self-sustaining, capable of spreading after initial establishment; and (iii) they minimize the need for non-renewable resources and offer enduring disease control in an environmentally sustainable manner [30,31,32]. Therefore, this study aimed to (i) isolate and identify indigenous marine *Bacillus* strains from mangrove ecosystems, (ii) assess their virulence against FAW neonates under laboratory conditions, (iii) evaluate the enzymatic activity of promising *Bacillus* isolates, (iv) identify the metabolic profile of potential strains using GC-MS, and (v) evaluate their colonization potential as endophytes in maize plants, along with their insecticidal activity against FAW.

## 2. Materials and Methods

### 2.1. Isolation of Bacterial Strains from the Mangrove Ecosystem

#### 2.1.1. Samples Collection

Fifteen samples from marine environments were collected from the mangrove ecosystem in Marsa Alam coastal area (25°52′01″ N 34°24′53″ E) in May 2020. The samples were collected from various biological sources, including sediment, rhizosphere, and endoshoot environments. Five sediment (S) samples were collected from around the mangrove tree (≥20 cm) at different depths (5, 10, 15, 20, and 25 cm), along with five rhizosphere (R) samples, and five endoshoot (ESh) samples. All samples were stored in sterile plastic bags and transported in an ice box to the laboratory.

#### 2.1.2. Samples Preparation and Bacterial Isolation

The endophytic shoot samples were surface sterilized by immersion in a 5% sodium hypochlorite solution for 3 min, followed by a 3% H_2_O_2_ solution for an additional 3 min, and then washed thoroughly with sterile 0.85% NaCl saline solution for at least 10 min. Afterwards, the samples were ground using a sterile mortar and pestle according to [33].

All samples were transferred to 45 mL of 0.85% NaCl and mixed vigorously at high speed with a vortex for 1 min. Then, 100 μL from each tenfold serial dilution of the resulting suspensions were applied to the surface of nutrient agar supplemented with 3% NaCl and nutrient agar with seawater media. Plates were incubated at 30 °C for 48 h, and bacterial colonies with distinct morphological characteristics were then purified by repeated streaking and kept in a glycerol solution at −80 °C until further use.

### 2.2. Antimicrobial Activity of Potent Isolates

The antagonistic activities of the isolates were evaluated against eight pathogens obtained from Cairo University Research Park (CURP) at the Faculty of Agriculture. The test bacteria included five Gram-positive bacteria, *Listeria monocytogenes* ATCC 13932, *Staphylococcus aureus* ATCC 25923, *Bacillus subtilis* ATCC 6633, *Bacillus cereus* ATCC 33018, and methicillin-resistant *Staphylococcus aureus* ATCC 43300; three Gram-negative bacteria, *Escherichia coli* ATCC 35218, *Pseudomonas aeruginosa* ATCC 9027, and *Salmonella typhi* ATCC 14028. All isolates were screened for antagonism in vitro against the eight tested microorganisms.

The bacterial isolates were enriched in their respective broth media and incubated at 30 °C for 24 h on a shaker incubator at 180 rpm. Pathogenic bacteria were inoculated onto Mueller Hinton agar, and 15 µL of each isolate was applied to the agar surface in a 3 mm diameter circle. Plates were then incubated for 24 h. All treatments were performed in triplicate. Inhibition zones around each isolate were measured, and isolates demonstrating strong antibacterial activity were selected [34].

### 2.3. Screening of Mangrove Bacterial Isolates Against Neonates’ Larvae of FAW

#### 2.3.1. Insect Culture

FAW cultures were obtained from the Department of Entomology at the Faculty of Agriculture, Cairo University, Egypt. Sexed pupae were placed in glass jars lined with paper towels until adult moths emerged. The incipient adult moths were transferred to a larger jar supplied with cotton wool soaked in a 10% sugar solution as a dietary supplement [35]. Eggs were collected daily and placed in new containers for hatching. Upon hatching, neonates were fed castor oil leaves, and mass-reared larvae were kept in sterilized plastic containers with a tightly sealed lid covered with tissue paper to prevent escape. After 7 days, larvae were separated into individual containers with a small piece of castor leaf to prevent cannibalism [36]. Experimental treatments were conducted using neonate FAW, and the culture was maintained under controlled conditions: 27 ± 1 °C, 70 ± 5% RH, and a 14L:10D photoperiod.

#### 2.3.2. Bioassay

Thirty-three bacterial isolates were selected based on their highly antagonistic effects against pathogenic bacteria. These isolates were cultured in nutrient broth supplemented with 3% NaCl on a shaker incubator at 30 °C for 48 h. Screening bioassays were conducted using leaf-dipping [37,38]. Castor leaves were dipped in approximately 10^7^ CFU/mL of each isolate for 20 s and then allowed to air dry for 30 min. Five replicates, each containing ten larvae, were placed individually in clean plastic cups (5 cm) for each isolate. Leaves dipped in water were used in the control group. The larvae were allowed to feed on the treated leaves for 24 h; then, the surviving larvae were transferred to clean, dry containers containing fresh untreated leaves.

Larval mortality was recorded daily for eight consecutive days post-treatment. All bioassays were conducted under controlled conditions of 27 ± 1 °C, 70 ± 5% RH, and a 14L:10D photoperiod. Mortality data were collected every 24 h until the eighth day after treatment, and the corrected percentage mortality was calculated using Abbott’s formula [39]. The mortality rate was determined using the following formula:$$Percent\;larval\;mortality=\frac{Number\;of\;dead\;larvae}{Total\;number\;of\;treated\;larvae}\;\times \;100$$

### 2.4. Enzymatic Activity of Promising Isolates

Based on a mortality rate of 80% or higher, three isolates (R39, Esh39, and Esh73) were screened for their enzymatic activity.

Catalase Activity: Catalase production was assessed by transferring a loopful of bacterial culture to a sterile glass slide and immediately adding a drop of 3% hydrogen peroxide. The presence of effervescence indicated catalase production [40].

Amylase Activity: Pure isolates were cultured for five days at 30 °C on starch agar medium, with the isolates spotted in the center of the plates. After incubation, a 2% iodine (KI) solution was poured into the plates [41]. The development of a clear zone around the colonies indicated amylase activity.

Protease Activity: Isolates were screened for their ability to generate protease on milk agar medium, which consisted of nutrient agar supplemented with 1% skimmed milk. After incubation at 30 °C for 2 days, proteolysis was assessed. The observation of a clear area around the colonies indicated protease activity [42].

Lipase Activity: The lipolytic activity of the isolates was evaluated using a nutrient medium supplemented with oil. After two days of incubation, lipolysis was assessed by adding a saturated copper sulfate solution. A blue-green color around the colonies indicated lipolytic activity [43].

Pectinase Activity: Pure isolates were cultured on pectin agar [44] at 30 °C for five days. After incubation, the plates were submerged in an aqueous solution containing 1% *w*/*v* hexadecyl-trimethyl-ammonium bromide and left to set at room temperature for 30 to 60 min. In contrast to the opaque non-hydrolyzed medium, a clear zone surrounding the colony indicated pectinolytic activity.

Nitrate Reduction: Stab cultures in a sloppy medium were treated with 0.2 mL of Griess-Ilosvary reagents I (0.8% sulfanilic acid in 5 N acetic acid) and II (0.5% naphthylamine in 5 N acetic acid) after 7 and 14 days. The presence of nitrate was indicated by a red coloration [40].

Phosphate Solubilization: The purified isolates were cultured as a spot in the center of [45] minimum agar medium plates at 30 °C for five days. Phosphate solubilization was detected by the formation of a clear area surrounding the colonies.

Hydrogen Sulfide Production: The production of H_2_S was detected by inserting lead acetate paper strips into the necks of culture tubes used for nitrate reduction determination. The blackening of the paper after the incubation indicates a positive result [46].

### 2.5. Metabolic Profiling of Potential Strains Using GC-MS

The three bacterial strains (R39, Esh39, and Esh73) showing promising biocontrol activity against *S. frugiperda* (with a mortality rate of ≥80%) were cultivated and incubated as previously described. The bacterial isolates were centrifuged at 5000× *g* for 15 min, then the cell-free supernatant was extracted using ethyl acetate. The chemical composition of the extracts was analyzed using a Trace GC-TSQ mass spectrometer (Thermo Scientific, Austin, TX, USA) equipped with a direct capillary column (TG–5MS) featuring dimensions of 30 m × 0.25 mm and a film thickness of 0.25 µm. The column oven temperature was initially maintained at 50 °C, increased to 250 °C at 5 °C/m, and held for 2 min; then further raised to 300 °C at a rate of 30 °C/m, and maintained for a duration of two minutes.

The MS transfer line and injector temperatures were held at 260 °C and 270 °C, respectively. The carrier gas was Helium at a 1 mL/min fixed flow rate. The solvent was set at 5 min, and automatically, 1 µL of the diluted samples was injected in split mode with GC mass. The collection of electron ionization (EI) mass spectra was performed in full scan mode at 70 eV over a range of m/z 50–650, with the ion source temperature held at 200 °C, as per [47]. The identification of components was achieved by comparing the mass spectra of components with of WILEY 09 and NIST 14 mass spectral databases.

### 2.6. Molecular Identification of the Promising Isolates

The three bacterial cultures were cultivated in LB broth medium for 24 h, then centrifuged at 10,000× *g* for 6 min. According to the manufacturer’s protocol, genomic DNA was extracted using the GeneJET Genomic DNA Purification kit (Cat. No. K0721). The measuring of the DNA concentration was performed using a NanoDrop spectrophotometer 2000, (Thermo Fisher Scientific, Munich, Germany). The DNA was kept at −20 °C. Afterwards, the 16s rRNA gene was amplified using general primer F-27 (5′-AGAGTTTGATCMTGGCTCAG-3′) and R1494 (5′-CTACGGYTACCTTGTTACGAC-3′) according to [48,49].

The PCR reactions were performed using a Bio-Rad T100 thermal cycler. Amplification conditions consisted of 30 cycles as follows: initial denaturation at 94 °C for 5 min, followed by denaturation at 94 °C for 1 min, annealing at 59 °C for 1 min, and extension at 72 °C for 2 min. The final extension was carried out at 72 °C for 10 min. The sequencing of PCR products corresponding to the partial 16S rRNA gene was conducted by Macrogen (Seoul, Republic of Korea). The sequences were compared against GenBank database similarity hits using BLASTn (https://blast.ncbi.nlm.nih.gov/Blast.cgi, 22 August 2024). Our isolates’ 16S rRNA gene sequences have been submitted to the NCBI GenBank database under accession numbers PQ197649 to PQ197651. The MEGA software (version 11) was used to construct a phylogenetic tree, employing the maximum composite likelihood method. The analysis included 30 sequences representing the closest matches identified in the GenBank database.

### 2.7. In Vitro Evaluation of the Three Promising Isolates (B. tequilensis, Bacillus spp. and B. subtilis) Under Greenhouse Conditions

#### 2.7.1. Inoculation Methods

Maize seeds (*Zea mays* L. single hybrid Giza 168) were obtained from the Faculty of Agriculture, Cairo University, and were surface sterilized following [27,28]. Sterilized peat moss (SAB, Syke, Germany) was used as the potting medium to ensure an environment free of contaminating microorganisms. Four inoculation methods (~10⁷ CFU/mL) were used to introduce the effective strains (*B. tequilensis* strain R39, *Bacillus* sp. strain Esh39, and *B. subtilis* strain Esh73) into the plants: foliar application (FA), seed treatment (ST), soil drenching (SD), and a combined method (FA + ST + SD). Each treatment was conducted in triplicate, with each replicate comprising three pots. (*n* = 9/isolate/treatment).

For seed and combination treatments, three sterilized seeds per pot were soaked for 2 h in cultures of each of the three *Bacillus* isolates, using 200 mL of *Bacillus* culture per treatment. Control seeds were soaked in distilled water for the same duration. Pots (25 cm in size, 750 g capacity) were filled with equal amounts of sterilized potting mixture, and seeds were sown at a rate of three seeds per pot, with three pots per treatment. The plants were maintained in a net house.

For foliar application, bacterial cultures were sprayed on the plants 15 days after emergence (DAE) until the leaves were adequately wet (~5 mL per plant). For soil drenching, the bacterial culture was injected near the roots of the 15-day-old plants at a rate of 5 mL per plant [8]. In combination treatment, seed treatment, soil drenching, and foliar application were applied. In all treatments, the respective controls were treated with distilled water.

#### 2.7.2. Measurement of Plant Height and Chlorophyll Content

Using a centimeter ruler, height measurements were taken from the base of the plant to its apex. Measurements were conducted at 20, 30, and 40 days after planting. Additionally, chlorophyll content was determined using a SPAD chlorophyll meter (Minolta, Osaka, Japan). Samples were taken from the third leaf, located two-thirds of the way up the plant.

#### 2.7.3. Impact of Inoculated Plants on *S. frugiperda*

Neonate larvae (<24 h old) were used for the bioassay. Leaf bits (2 × 2 cm) were collected from plants 20 days after inoculation (DAI). Each treatment was replicated three times, with 10 larvae per replicate. Leaf bits were replaced daily to prevent food shortages and maintain moisture. Larval mortality was recorded daily for six days, and the mortality rates were calculated for all treatments and the control group.

### 2.8. Statistical Analysis

Data were analyzed using SPSS (V.22). The normality of continuous variables was assessed with the Shapiro–Wilk and Kolmogorov–Smirnov tests. An ANOVA was conducted to compare treatment and control groups, followed by Tukey’s post hoc pairwise analysis. A *p*-value of <0.05 was considered statistically significant. SigmaPlot (V.12.0) was used for graphing and additional analysis.

## 3. Results

### 3.1. Isolation and Antagonistic Activity of Mangrove Bacterial Isolates

As summarized in Figure 1, the distribution of bacterial isolates varied across different zones of the mangrove trees, sediments, and the used media. The greatest number of isolates was derived from the endoshoot sphere, followed by the rhizosphere. Notably, the nutrient agar medium supplemented with 3% NaCl yielded more isolates than the seawater agar medium. In total, 200 bacterial isolates were evaluated for their antibacterial activity against eight pathogens. Among these, 33 isolates exhibited significant inhibition zones against the tested pathogens, as shown in Table 1. Strains S48, R4, R64, and Esh2 demonstrated high efficacy against all Gram+ and Gram- pathogens. Strain R64 exhibited the largest zones of inhibition, 23 mm and 30 mm against Gram+ pathogens except for *L. monocytogenes* (17 mm), and 25 mm and 12 mm against Gram- pathogens except *S. typhi* (2 mm). In total, 81% of the isolates exhibited inhibitory effects against *P. aeruginosa,* while 78% showed inhibition against *Escherichia coli*, *Salmonella typhi*, and *Bacillus cereus,* and only 42% showed inhibition against *Listeria monocytogenes*.

### 3.2. Bioassay of Mangrove Bacterial Isolates Against Neonates of S. frugiperda

Screening of thirty-three bacterial isolates exhibited varying levels of virulence against neonates of *S. frugiperda.* As shown in Figure 2, Esh73 caused the maximum significant mortality rate (80%) after 24 h of treatment, followed by Esh39 (74%), Esh45 (70%), and Esh38 (68%) after 2 days of treatment, compared to the control [*F* = 7.92; *df* = 34; *p* = 0.0001]. After 8 days of treatment, Esh39, R39, and Esh73 exhibited higher larval mortality, i.e., 83.54%, 83.36%, and 82%, respectively [*F* = 1.72; *df* = 34; *p* = 0.016], and thus were further investigated for enzymatic activity, metabolic profiling, and ability to colonize in maize plants.

### 3.3. Enzymatic Activity of the Three Promising Isolates

The three promising strains produced amylase and catalase and reduced nitrate. Strain R39 produced pectinase and weakly solubilized phosphate, while strain Esh73 produced lipase. However, none of the three strains were able to produce protease and H_2_S.

### 3.4. Molecular Identification of the Three Promising Isolates

The BLAST analysis of the 16S rDNA sequences from strains R39, Esh39, and Esh73 revealed a 100% similarity to *B. tequilensis* R39, *Bacillus* spp. Esh 39 and *B. subtilis* Esh73, respectively, as shown in Figure 3. The 16srRNA sequences have been submitted to GenBank and assigned accession numbers ranging from PQ197649 to PQ197651. The neighbor-joining phylogenetic tree constructed for the bacterial isolates demonstrated significant phylogenetic relatedness (Figure 3).

### 3.5. Metabolic Profiling of Bacillus Strains Using GC-MC

The chemical compounds of *B. tequilensis* are shown in Table 2 and Appendix A. The major constituents in this strain were 1-Docosene (9.69%), Diisooctyl phthalate (9.05%), 1-Nonadecene (8.09%), Nonacos-1-ene (6.77%), and Hexadecanoic acid (5.12%). Additionally, Table 2 and Appendix A show that the major components in *Bacillus* spp were 9,12-Octadecadienoic acid (z,z)-, 2,3-bis[(trimethylsilyl)oxy]propyl ester (20.60%), Arabinitol, pentaacetate (8.16%), Diisooctyl phthalate (8.11%), and Nonacos-1-ene (7.18%). As for *B. subtilis* (Table 2 and Appendix A), the main components were Diisooctyl phthalate (9.56%), 1-Docosene (7.75%), 9,12-Octadecadienoic acid (z,z)-, 2,3-bis[(trimethylsilyl)oxy]propylester (6.96%), 1-Docosene (6.50%), 3,4-Dihydro-2h-1,5-(3”-t-butyl) benzodioxepine (5.77%), and Nonacos-1-ene (5.33%).

### 3.6. Efficacy of the Three Plant-Colonizing Bacterial Isolates Against S. frugiperda 

The efficacy of the three bacterial strains (*B. tequilensis*, *Bacillus* sp., and *B. subtilis*) as endophytic biocontrol agents in maize plants against neonates of the Fall Armyworm (*S. frugiperda*) is shown in Table 3. As to seed treatment, *Bacillus* sp. and *B. subtilis* strains caused a significantly higher mortality rate (53.33%) than the control. For foliar application, *Bacillus* sp. recorded the highest mortality rate (65%), followed by *B. tequilensis* (60%), both significantly exceeding the control. For soil drenching, *B. subtilis* achieved the highest mortality rate (56.67%). When all inoculation methods were combined, *B. subtilis* caused the highest mortality (60%) compared to the control.

### 3.7. Effectiveness of the Three Plant-Colonizing Bacterial Isolates in Promoting Plant Height and Chlorophyll Content

The variation in plant height with different application methods is summarized in Table 4 and Appendix A. Seed treatment and soil drenching of the three strains caused significant effects on plant height at 40 days [F = 12.18; *p* = 0.001, and F = 8.83; *p* = 0.001, respectively]. Based on average plant heights, *Bacillus* sp. showed the greatest increase, followed by *B. tequilensis* and *B. subtilis*. For foliar application, no significant differences in plant height were observed at 20, 30, and 40 days across all strains [F = 2.38; *p* = 0.058, F = 2.08; *p* = 0.090, and F = 2.30; *p* = 0.110, respectively]. On the other hand, the combination of the three methods caused a significant increase in plant height at 20, 30, and 40 days [F = 5.77; *p* = 0.001, F = 4.97; *p* = 0.001, and F = 8.05; *p* = 0.002, respectively]. The variation in chlorophyll content (mg/m^2^) in maize plants is shown in Table 5 and Appendix A. Seed treatment and foliar application of the three strains caused a significant increase in chlorophyll content at 40 days [F = 13.27; *p* = 0.001, and F = 6.61; *p* = 0.001, respectively]. As for seed treatment, *B. subtilis* led to the highest chlorophyll levels, followed by *Bacillus* sp., and *B. tequilensis*. In foliar application, *B. tequilensis* led to the highest chlorophyll content, followed by *Bacillus* sp., and *B. subtilis*. On the other hand, no significant differences in chlorophyll content were observed at both 30 and 40 days with soil drenching [F = 2.69; *p* = 0.050, and F = 2.26; *p* = 0.086, respectively]. However, when the three application methods were combined, a significant increase in chlorophyll content was noted at 40 days [F = 5.65; *p* = 0.002].

## 4. Discussion

The FAW managing strategies have traditionally relied on chemical pesticides. Nonetheless, these pesticides pose significant environmental and health risks, emphasizing the urgent need for eco-friendly alternatives in pest control. Microbial biocontrol agents, particularly *Bacillus species*, have emerged as leading solutions, offering effective pest management while promoting plant growth and supporting sustainable agriculture [67]. The *Bacillus* genus is characterized by a remarkable diversity in metabolism and genetics, allowing these bacteria to thrive across diverse environments, including marine and terrestrial ecosystems [68]. Due to their ability to form durable endospores, *Bacillus*-based biocontrol products are highly stable, ensuring effectiveness and long-term persistence in the environment [69,70].

In this study, 200 *Bacillus* strains were isolated from the mangrove ecosystem along Egypt’s Marsa Alam coast. The isolates were primarily sourced from mangrove tree shoots (77 isolates, 38.5%), followed by the rhizosphere (64 isolates, 32%) and sediment (59 isolates, 29.5%). Of these, 33 *Bacillus* isolates demonstrated antagonist activities against a range of both Gram-positive and Gram-negative bacteria, underscoring mangrove forests as rich reservoirs of biological diversity and effective bioactive compounds [71]. Additionally, the dynamic salinity conditions and abundant organic matter in mangrove ecosystems foster microbial adaptations, encouraging *Bacillus* species to produce a broad spectrum of bioactive compounds, including effective biocontrol agents that induce mortality in different insects’ orders [72,73]. The bioassay screening of 33 Bacillus isolates against *S. frugiperda* neonates revealed that the *Bacillus subtilis* Esh73 exhibited the highest larval mortality (80%) within two days post-treatment. After eight days, *Bacillus sp.* Esh39, *B. tequilensis* R39, and *B. subtilis* Esh73 maintained high larval mortality rates, respectively, indicating a potent insecticidal activity against *S. frugiperda* larvae. This high lethality is likely attributed to the bioactive secondary metabolites produced by these *Bacillus* isolates [74]. Previous studies have shown that *B. subtilis* produces bioactive lipopeptide metabolites that have been used to control *Spodoptera littoralis* [75] *Drosophila melanogaster* [76], *Culex quinquefasciatus* [77], *Anopheles stephensi* [78], and *Aedes aegypti* [79]. However, [80] reported that two *Bacillus* strains, including *B. subtilis* (CQ2) and *B. tequilensis* (CQ3), had insecticidal effects on *Cimbex quadrimaculatus* (Hymenoptera: Cimbicidae). These effects ranged from 58 to 100% and reached 100% in *B. subtilis* (CQ2) in the third instar larvae after 10 days post-treatment of 1.89 × 10^9^ cfu/mL bacterial concentration under laboratory conditions. In addition, GC-MS analysis was conducted to determine the major components in the crude extracts. The results showed that *B. tequilensis* R39 contains several insecticidal compounds, including 1-Docosene, 1-Nonadecene, and Hexadecanoic acid while *Bacillus sp.* Esh39 and *B. subtilis* Esh73 contained 9,12-Octadecadienoic acid (z,z)-, 2,3-bis[(trimethylsilyl)oxy]propyl ester, a compound known for its insecticidal properties. *Bacillus* species are generally renowned for producing secondary metabolites with broad-spectrum bioactivity against pests and pathogens and plant growth-promoting effects [81,82].

To manage the resistance development in *S. frugiperda* and enhance the effectiveness of entomopathogenic strains in field applications, colonization of these pathogens within plants is valuable. This approach may improve nutrient uptake, increase tolerance to abiotic stresses, enhance plant growth, and boost plant resistance to herbivores [83,84]. In this study, three entomopathogenic strains—*B. tequilensis* R39, *Bacillus* sp. Esh39, and *B. subtilis* Esh73—were inoculated into maize plants using four techniques: foliar application, seed treatment, soil drenching, and a combination of the three techniques. Bioassays against FAW neonates indicated that foliar application achieved the highest mortality rates, with 65% for *Bacillus* sp. Esh39 and 60% for *B. tequilensis* R39, outperforming control plants under similar conditions. However, this high mortality did not correlate with enhanced plant height or chlorophyll content. The seed treatment method demonstrated a comparatively lower mortality rate among *S. frugiperda* neonates than the other inoculation methods. However, 40 days after planting, *Bacillus* sp., *B. tequilensis*, and *B. subtilis* enhanced plant height, achieving 115, 110, and 106 cm, respectively, compared to the control (88 ± 2 cm). Similarly, *B. subtilis*, *Bacillus* sp., and *B. tequilensis* notably increased chlorophyll content by approximately 1.58, 1.52, and 1.51 times, respectively, compared to the control. Consistent with these results, prior findings reported that *Bacillus* species, including *B. tequilensis* [85], *B. subtilis* [86], and *B. thuringiensis* [27], exhibit strong plant-colonization abilities and bioactivity against various pests and pathogens. In previous studies, the *B. subtilis* strain isolated from soil promoted plant growth [87] and had potentially inhibited the biological activity of *Cnaphalocrocis medinalis* [73]. The bacterial endophyte *B. tequilensis* (PBE1) was proven to be the most effective control agent against *Fusarium oxysporum* for tomato wilt disease management and also showed efficient plant growth-promoting effects by producing indole acetic acid (IAA), hydroxymate-type siderophore, along with phosphate solubilizing ability [88]. This study provides significant insights into the bacterial diversity and functional roles within the mangrove ecosystem, highlighting the isolation and characterization of numerous *Bacillus* strains associated with mangrove plants. These isolates demonstrated a wide range of biological activities, encompassing antibacterial and insecticidal features, particularly against *S. frugiperda* neonates. Additionally, the isolates showed capabilities in producing hydrolytic enzymes—such as amylase, catalase, lipase, and pectinase—and in solubilizing phosphate, reducing nitrate, and enhancing plant growth. Understanding the microbial diversity and ecosystem functions within mangroves is essential for advancing sustainable agricultural practices and informing conservation policies. Consequently, establishing native *Bacillus* strains as endophytes in maize plants, with inherent insecticidal properties, presents a promising alternative for *S. frugiperda* management, potentially reducing the need for genetically modified plants.

## 5. Conclusions

The findings of this study highlight the importance of exploring microbial diversity in unique ecosystems like mangroves, where bioactive compounds with multifunctional properties are produced. The study highlights the potential of mangrove-associated Bacillus strains as eco-friendly biocontrol agents, offering effective management of FAW while promoting sustainable agricultural practices. *B. tequilensis* R39, *Bacillus sp.* Esh39, and *B. subtilis* Esh73 exhibited strong insecticidal activity and plant growth-promoting capabilities. The ability of these strains to colonize maize plants and enhance their growth further emphasizes their convenience as alternatives to conventional chemical pesticides and genetically modified crops. However, future research should focus on scaling field applications, optimizing inoculation methods, and evaluating long-term environmental impacts to integrate these native *Bacillus* strains into sustainable pest management strategies effectively.

## Figures and Tables

**Figure 1 biology-13-01057-f001:**
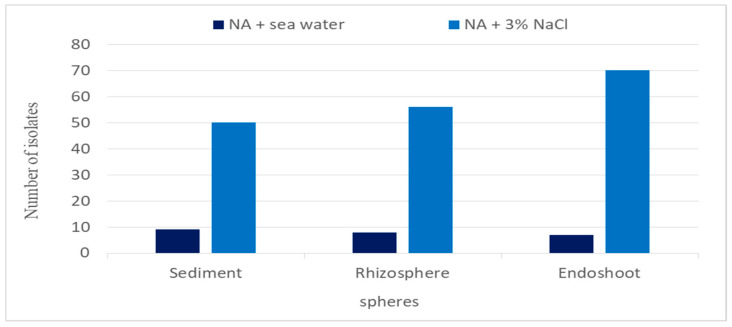
Total isolates from different mangrove spheres and sediments using two isolation media.

**Figure 2 biology-13-01057-f002:**
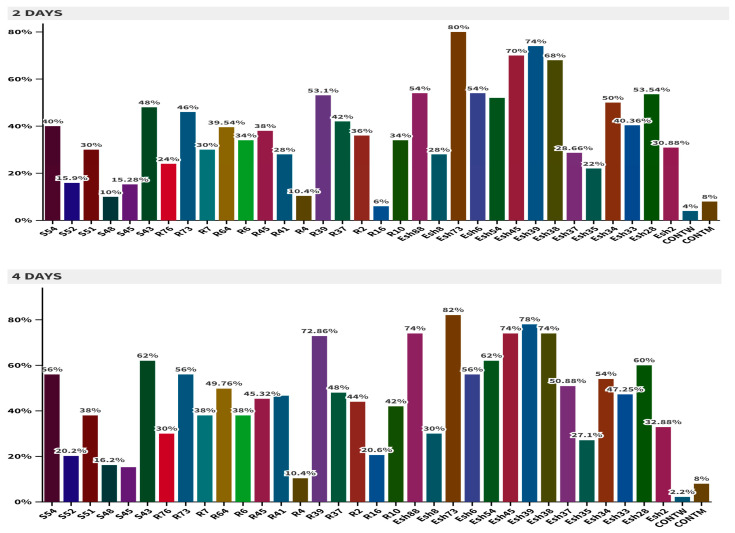
Percentage mortality of *S. frugiperda* neonates by thirty-three bacterial isolates.

**Figure 3 biology-13-01057-f003:**
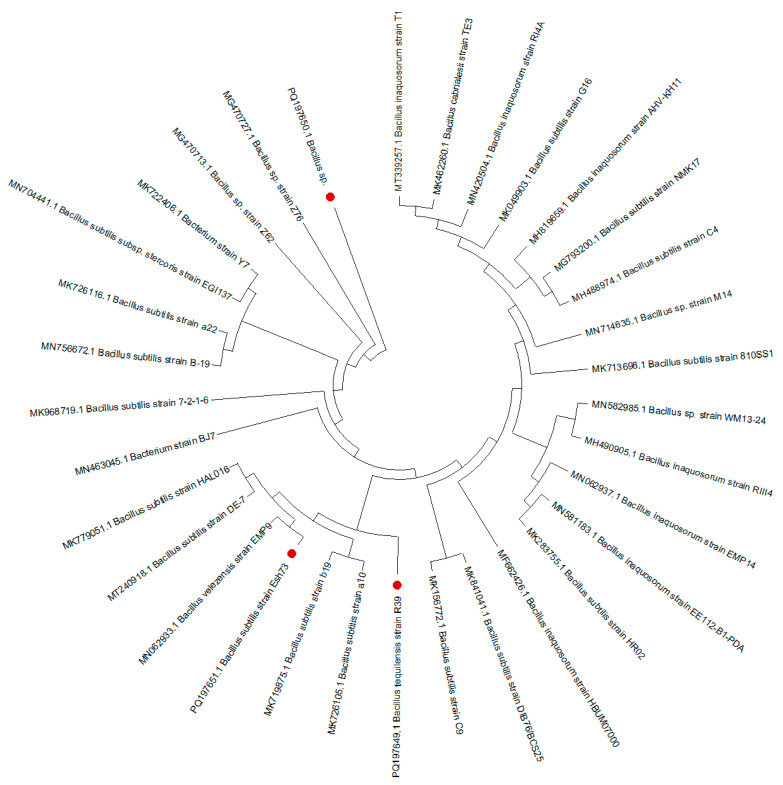
Neighbor-joining phylogenetic tree based on 16S rRNA gene sequences of three bacterial isolates (highlighted with red circles) and closest matches from the NCBI database.

**Table 1 biology-13-01057-t001:** Antagonistic activity of bacterial isolates against pathogenic bacteria.

Isolates	Inhibition Zones (mm)							
	Gram^−^ Bacteria			Gram^+^ Bacteria				
	*Sal. typhi*	*E. coli*	*Ps. aeruginosa*	*L. monocytogenes*	*St. aureus*	*St. aureus* (MRSA)	*B. cereus*	*B. subtilis*
S43	-	5	11	16	8	15	8	14
S45	4	6	12	-	-	-	5	-
S48	5	7	7	5	30	24	7	15
S51	4	4	5	-	8	8	8	-
S52	13	7	4	-	-	-	7	-
S54	5	10	6	13	-	-	7	23
R2	10	5	3	-	6	-	6	-
R4	6	3	7	7	2	2	1	5
R6	6	15	5	6	6	-	4	-
R7	2	2	2	-	7	4	8	11
R10	7	25	9	-	8	-	6	20
R16	-	-	2	-	23	25	12	30
R37	-	-	-	-	10	10	6	-
R39	-	-	-	-	9	8	23	-
R41	-	2	-	-	22	21	8	-
R45	2	4	-	-	11	12	6	5
R64	2	27	1 2	17	30	30	23	30
R73	-	-	3	8	6	2	12	10
R76	5	5	-	-	13	5	17	23
Esh2	11	10	8	14	17	19	2	27
Esh6	2	1	-	-	-	12	12	13
Esh8	4	3	4	-	-	-	-	-
Esh28	3	5	2	-	-	-	-	-
Esh33	5	-	5	13	3	2	6	21
Esh34	16	-	12	11	4	7	11	14
Esh35	6	5	10	-	2	2	4	9
Esh37	-	-	4	20	13	8	8	15
Esh38	4	4	5	14	15	10	6	-
Esh39	6	5	11	13	-	-	-	14
Esh45	10	9	5	-	4	2	-	-
Esh54	6	4	2	-	-	-	-	-
Esh73	10	4	6	16	2	2	-	-
Esh88	5	4	4	-	12	6	-	-

**Table 2 biology-13-01057-t002:** Major compounds in the secondary metabolites of three bacterial isolates as identified by GC-MS analysis.

Isolates	RT (min)	Compound Name	Area %	MF	Molecular Formula	Biological Activities	References
*Bacillus tequilensis*	26.42	Hexadecanoic acid	5.12	789	C_16_H_32_O_2_	Antimicrobial, insecticidal, antioxidant activity and plant growth promoters	[50,51,52,53,54,55]
	27.14	1-Nonadecene	8.09	952	C_19_H_38_	Antimicrobial, antifungal and insecticidal activity	[56,57,58]
	30.75	1-Docosene	9.69	933	C_22_H_44_	Antibacterial and insecticidal	[58,59,60,61]
	34.07	Nonacos-1-ene	6.77	933	C_29_H_58_	Antibacterial	[62]
	35.77	Diisooctyl phthalate	9.05	952	C_24_H_38_O_4_	Antimicrobial and antifouling effect	[63]
*Bacillus* spp.	30.75	Nonacos-1-ene	7.18	925	C_29_H_58_	Antibacterial	[62]
	34.07	2-Hexadecanol	5.20	829	C_16_H_34_O	Antibacterial	[62]
	35.77	Diisooctyl phthalate	8.11	947	C_24_H_38_O_4_	Antimicrobial and antifouling effect	[63]
	42.87	9,12-Octadecadienoic acid (z,z)-,2,3bis[(trimethylsilyl)oxy]prpyl ester	20.60	716	C_27_H_54_O_4_Si_2_	Insecticide activity, antimicrobial and antioxidant	[53,64]
	43.22	Arabinitol, pentaacetate	8.16	691	C_15_H_22_O_10_	Antimicrobial,	[65]
*Bacillus subtilis*	16.63	3,4-Dihydro-2h-1,5-(3″-t-butyl)benzodioxepine	5.77	960	C_13_H_18_O_2_	Antioxidant	[66]
	27.14	1-Docosene	6.50	951	C_22_H_44_	Antibacterial and insecticidal	[59,60,61]
	35.76	Diisooctyl phthalate	9.56	955	C_24_H_38_O_4_	Antimicrobial, antifungal and antifouling effect	[63]
	42.85	9,12-Octadecadienoic acid (z,z)-,2,3bis[(trimethylsilyl)oxy]propyl ester	6.96	719	C_27_H_54_O_4_Si_2_	Insecticide activity, antimicrobial and antioxidant	[53,64]

**Table 3 biology-13-01057-t003:** Mean cumulative mortality (±SD) of *Spodoptera frugiperda* neonates fed inoculated maize leaves using various inoculation methods: seed treatment (ST), soil drenching (SD), foliar application (FA), and combined method (ST + SD + FA) at 14- and 21-days post-inoculation.

	Treatments	1 Day	2 Days	3 Days	4 Days	5 Days	6 Days
Seed Treatment	*Bacillus tequilensis*	13.33 ± 11.55 a	20 ± 0 ab	40 ± 20 a	46.67 ± 30.55 ab	51 ± 25.36 a	51 ± 25.36 a
	*Bacillus* spp.	0 a	0 ab	20 ± 0 ab	53.33 ± 23.09 a	53.33 ± 23.09 a	53.33 ± 23.09 a
	*Bacillus subtilis*	6.66 ± 11.55 a	13.33 ± 11.55	26.67 ± 11.55 ab	33.33 ± 11.55 ab	53.33 ± 23.09 a	53.33 ± 23.09 a
	Control M	0 a	4 ± 8.94 b	4 ± 8.94 b	8 ± 17.89 b	8 ± 17.89 b	8 ± 17.89 b
	Control W	4 ± 8.94 a	4 ± 8.94 b	4 ± 8.94 b	8 ± 10.95 b	8 ± 10.95 b	8 ± 10.95 b
	*p*-Value	0.172	0.006	0.002	0.012	0.003	0.003
	*F*-Value (*df*)	1.79 (5)	4.98 (5)	6.26 (5)	4.24 (5)	5.84 (5)	5.84 (5)
Foliar Application	*Bacillus tequilensis*	13.33 ± 11.55 b	33.33 ± 11.55 a	46.67 ± 23.09 a	53.33 ± 11.55 a	60 ± 0 a	60 ± 0 a
	*Bacillus* spp.	40 ± 0 a	46.67 ± 11.55 a	53.33 ± 11.55 a	60 ± 0 a	65 ± 8.66 a	65 ± 8.66 a
	*Bacillus subtilis*	6.67 ± 11.55 b	26.67 ± 11.55 ab	33.33 ± 11.55 ab	40 ± 0 a	56.67 ± 5.77 a	56.67 ± 5.77 a
	Control M	0 b	4 ± 8.94 b	4 ± 8.94 b	8 ± 17.89 b	8 ± 17.89 b	8 ± 17.89 b
	Control W	4 ± 8.94 b	4 ± 8.94 b	4 ± 8.94 b	8 ± 10.95 b	8 ± 10.95 b	8 ± 10.95 b
	*p*-Value	0.001	0.001	0.001	0.001	0.001	0.001
	*F*-Value (*df*)	9.89 (5)	10.30 (5)	13.70 (5)	18.96 (5)	24.43 (5)	24.43 (5)
Soil Drenching	*Bacillus tequilensis*	20 ± 20 a	33.33 ± 11.55 a	33.33 ± 11.55 a	33.33 ± 11.55 ab	40 ± 20 ab	40 ± 20 ab
	*Bacillus* spp.	20 ± 0 a	13.67 ± 22.85 a	40 ± 0 a	40 ± 0 ab	53.33 ± 11.55 a	53.33 ± 11.55 a
	*Bacillus subtilis*	13.33 ± 11.55 a	53.33 ± 23.09 a	53.33 ± 23.09 a	53.33 ± 23.09 a	56.67 ± 20.82 a	56.67 ± 20.82 a
	Control M	0 a	4 ± 8.94 a	4 ± 8.94 b	8 ± 17.89 b	8 ± 17.89 b	8 ± 17.89 b
	Control W	4 ± 8.94 a	4 ± 8.94 a	4 ± 8.94 b	8 ± 10.95 b	8 ± 10.95 b	8 ± 10.95 b
	*p*-Value	0.069	0.281	0.001	0.003	0.001	0.001
	*F*-Value (*df*)	2.57 (5)	1.39 (5)	10.85 (5)	5.89 (5)	8.94 (5)	8.94 (5)
Combination	*Bacillus tequilensis*	0 b	13.33 ± 23.09 ab	20 ± 20 ab	33.33 ± 11.55 ab	43.33 ± 5.77 a	43.33 ± 5.77 a
	*Bacillus* spp.	13.33 ± 11.55 ab	13.33 ± 11.55 ab	13.33 ± 11.55 ab	13.33 ± 11.55 ab	26.67 ± 11.55 ab	53.33 ± 11.55 a
	*Bacillus subtilis*	26.66 ± 11.55 a	33.33 ± 11.55 a	40 ± 0 a	46.67 ± 11.55 a	53.33 ± 11.55 a	60 ± 0 a
	Control M	0 b	4 ± 8.94 b	4 ± 8.94 b	8 ± 17.89 b	8 ± 17.89 b	8 ± 17.89 b
	Control W	4 ± 8.94 b	4 ± 8.94 b	4 ± 8.94 b	8 ± 10.95 b	8 ± 10.95 b	8 ± 10.95 b
	*p*-Value	0.007	0.037	0.003	0.013	0.001	0.001
	*F*-Value (*df*)	4.90 (5)	3.14 (5)	5.79 (5)	4.20 (5)	9.32 (5)	13.48 (5)

Means, followed by the same letters within columns are not significantly different (*p* < 0.05).

**Table 4 biology-13-01057-t004:** Plant height (cm) of maize as affected by three plant-colonizing bacterial isolates using different inoculation methods: seed treatment (ST), soil drench (SD), foliar application (FA), and combined methods (ST + SD + FA) at 20-, 30- and 40-days post-inoculation.

		20 Days	30 Days	40 Days
Seed treatment	*Bacillus tequilensis*	39.06 ± 7.72 a	56.78 ± 13 a	110 ± 11.36 ab
	*Bacillus* spp.	42.56 ± 7.15 a	61.67 ± 12.06 a	115 ± 12.77 a
	*Bacillus subtilis*	39 ± 7.79 a	63 ± 12.06 a	106 ± 5 ab
	Control M	35.67 ± 1.52 a	43.33 ± 3.79 a	74.3 ± 4.04 c
	Control W	33.5 ± 4.93 a	46.83 ± 9.99 a	88 ± 2.65 bc
	*p*-Value	0.322	0.047	0.001
	*F*-Value (*df*)	1.21 (5)	2.50 (5)	12.18 (5)
Foliar application	*Bacillus tequilensis*	31.61 ± 6.61 ab	47.89 ± 13.91 a	81.67 ± 7.63 a
	*Bacillus* spp.	29.38 ± 7.23 b	46.5 ± 13.4 a	86.67 ± 2.08 a
	*Bacillus subtilis*	34.71 ± 8.03 ab	53.14 ± 13.32 a	87.67 ± 6.8 a
	Control M	35.67 ± 1.52 ab	43.33 ± 3.79 a	74.33 ± 4.04 a
	Control W	33.5 ± 4.93 ab	46.83 ± 9.99 a	88 ± 2.64 a
	*p*-Value	0.058	0.090	0.110
	*F*-Value (*df*)	2.38 (5)	2.08 (5)	2.30 (5)
Soil drenching	*Bacillus tequilensis*	23.67 ± 3.31 b	35 ± 4.58 a	72.33 ± 4.04 c
	*Bacillus* spp.	29.56 ± 7.65 ab	37.89 ± 11.6 a	88.33 ± 8.5 a
	*Bacillus subtilis*	29.22 ± 6.62 ab	39.33 ± 10.68 a	85.67 ± 4.04 ab
	Control M	35.67 ± 1.52 a	43.33 ± 3.78 a	74.33 ± 4.04 bc
	Control W	33.5 ± 4.93 a	46.83 ± 9.98 a	88 ± 2.64 a
	*p*-Value	0.016	0.236	0.001
	*F*-Value (*df*)	3.24 (5)	1.43 (5)	8.83 (5)
Combination	*Bacillus tequilensis*	34.56 ± 3.5 b	57.22 ± 6.3 ab	91.33 ± 14.05 abc
	*Bacillus* spp.	35.89 ± 2.02 ab	53.89 ± 9.1 ab	100.33 ± 10.5 ab
	*Bacillus subtilis*	39.56 ± 2.35 a	62.56 ± 6.4 a	114.67 ± 4.73 a
	Control M	35.67 ± 1.52 ab	43.33 ± 3.78 b	74.33 ± 4.04 c
	Control W	33.5 ± 4.93 b	46.83 ± 9.98 b	88 ± 2.65 bc
	*p*-Value	0.001	0.001	0.002
	*F*-Value (*df*)	5.77 (5)	4.97 (5)	8.05 (5)

Means followed by the same letters within columns are not significantly different (*p* < 0.05).

**Table 5 biology-13-01057-t005:** Chlorophyll content (mg/m^2^) in maize plant as affected by three plant-colonizing bacterial isolates using different inoculation methods: seed treatment (ST), soil drench (SD), foliar application (FA), and combined method (ST + SD + FA) at 30- and 40-days post-inoculation.

		30 Days	40 Days
Seed Treatment	*Bacillus tequilensis*	41.56 ± 4.63 a	40.5 ± 2.72 a
	*Bacillus* spp.	42.76 ± 2.28 a	40.78 ± 2.07 a
	*Bacillus subtilis*	42.98 ± 2.31 a	42.42 ± 3.82 a
	Control M	39.3 ± 1.04 a	28.6 ± 5.3 b
	Control W	39.77 ± 3.61 a	26.75 ± 6.41 b
	*p*-Value	0.413	0.001
	*F*-Value (*df*)	1.05 (5)	13.27 (5)
Foliar Application	*Bacillus tequilensis*	40.76 ± 3.88 a	40.64 ± 2.23 a
	*Bacillus* spp.	43.48 ± 2.85 a	38.68 ± 4.25 a
	*Bacillus subtilis*	38.82 ± 5.1 a	37.38 ± 2.82 ab
	Control M	39.3 ± 1.04 a	28.6 ± 5.3 bc
	Control W	35.85 ± 6.96 a	29 ± 7.09 c
	*p*-Value	0.211	0.001
	*F*-Value (*df*)	1.57 (5)	6.61 (5)
Soil Drenching	*Bacillus tequilensis*	24.7 ± 5.62 a	21.3 ± 2.21 a
	*Bacillus* spp.	32.1 ± 10.21 a	29.46 ± 8.74 a
	*Bacillus subtilis*	37.22 ± 5.6 a	32.1 ± 6.68 a
	Control M	39.3 ± 1.04 a	28.6 ± 5.3 a
	Control W	39.77 ± 3.61 a	26.75 ± 6.41 a
	*p*-Value	0.050	0.086
	*F*-Value (*df*)	2.69 (5)	2.26 (5)
Combination	*Bacillus tequilensis*	42.78 ± 2.15 a	36.44 ± 2.84 ab
	*Bacillus* spp.	43.34 ± 2.38 a	35.86 ± 1.43 ab
	*Bacillus subtilis*	40.74 ± 1.37 a	38.94 ± 4.7 a
	Control M	39.3 ± 1.04 a	28.6 ± 5.3 c
	Control W	40.3 ± 1.9 a	31.18 ± 2.69 bc
	*p*-Value	0.166	0.002
	*F*-Value (*df*)	1.75 (5)	5.65 (5)

Means within columns followed by the same letters within columns are not significantly different (*p* < 0.05).

## Data Availability

The datasets generated during the current study are available from the corresponding author upon reasonable request.

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
