# Peer review of "Isolation and Characterization of Bacillus Strains from Egyptian Mangroves: Exploring Their Endophytic Potential in Maize for Biological Control of Spodoptera frugiperda"

_biology, 2024, doi:10.3390/biology13121057_

Round 1
Reviewer 1 Report
Comments and Suggestions for Authors
The Manuscript entitled “Isolation and Characterization of Bacillus Strains from Egyptian Mangroves: Exploring Their Endophytic Potential in Maize for Biological Control of Spodoptera frugiperda” is suitable for acceptance after revision
· Required revision as per comments and suggestions
Revise as comments below
1. Key words should be A-Z pattern. The key words should be different words already present in the title of manuscript, it should be focused on topic.
2. Page 2, line 54: Add and before China.
3. Page 4, lines 156-157: In material and methods section the use of the leaf-dip method is appropriate for evaluating the effects of bacterial isolates on larvae. However, it is important to provide references to support the choice of this technique. I recommend citing relevant literature, such as Nauen et al. (2008) and Cuthbertson et al. (2009), to strengthen the methodological context and acknowledge prior work.
4. Page 4, lines 157: Why the only one bacterial concentration of 107 CFU/mL was chosen.
5. Add footnotes at the end of Table 3, 4 & 5 to enhance clarity and ensure the data is more accessible and reproducible
6. Revise figures with high quality if possible
7. The genus name is not always etalic. Following correct nomenclature norms in scientific writing requires italicizing both the genus and species names.
Author Response
Reviewer 1
The Manuscript entitled “Isolation and Characterization of Bacillus Strains from Egyptian Mangroves: Exploring Their Endophytic Potential in Maize for Biological Control of Spodoptera frugiperda” is suitable for acceptance after revision
Required revision as per comments and suggestions
Revise as comments below
- Key words should be A-Z pattern. The key words should be different words already present in the title of manuscript, it should be focused on topic.
Reply: Done
- Page 2, line 54: Add and before China.
Reply: Done
- Page 4, lines 156-157: In material and methods section the use of the leaf-dip method is appropriate for evaluating the effects of bacterial isolates on larvae. However, it is important to provide references to support the choice of this technique. I recommend citing relevant literature, such as Nauen et al. (2008) and Cuthbertson et al. (2009), to strengthen the methodological context and acknowledge prior work.
Reply: Done
- Page 4, lines 157: Why the only one bacterial concentration of 107 CFU/mL was chosen.
Reply: We used the McFarland standard of 0.5, which is equivalent to 107 CFU/mL. In order to increase the larval mortality, we are working on the different factors, including bacterial inoculum size now, which will maximize and optimize the larval death rate.
- Add footnotes at the end of Table 3, 4 & 5 to enhance clarity and ensure the data is more accessible and reproducible
Reply: Done
- Revise figures with high quality if possible
Reply: Done
- The genus name is not always etalic. Following correct nomenclature norms in scientific writing requires italicizing both the genus and species names.
Reply: Done
Reviewer 2 Report
Comments and Suggestions for Authors
These are my main comments on the manuscript (Biology-3275353) entitled “Isolation and Characterization of Bacillus Strains from Egyptian Mangroves: Exploring Their Endophytic Potential in Maize for Biological Control of Spodoptera frugiperda”. This work investigates the colonization potential of indigenous marine Bacillus strains as endophytes in maize plants and insecticidal activity against S. frugiperda. However, details about introduction, materials and methods, and results section are needed. Following substantial revisions should be incorporated in the manuscript prior to acceptance.
L.17: …damaging maize crops and…
L.19: …of bacteria found in…
L.22: …particularly Bacillus…
L.23: …R39 caused significant mortality in laboratory…
Ls.44-45: Keywords should be in alphabetic order. Also, keywords serve to widen the opportunity to be retrieved from a database. To put words that already are into title and abstracts makes KW not useful. Please choose terms that are neither in the title nor in abstract.
Ls.48-50: Summarize this sentence.
L.51: … historically been a problem in the…
Ls.53-54: …China and ? Incomplete sentence, revise.
L.54: Sentence starting “The FAW…”.
L.57: Change “produce” by “products”.
Ls.57 and 62: Please, abbreviated for FAW. Correct in all manuscript.
L.63: Delete “the”.
L.73: The word “Bt” should be in italic.
L.88: Again, the word “Bacillus” should be in italic.
L.101: Also, a hypothesis for this study is needed.
L.131: …ATCC 43300, 3…
L.132: …ATCC 9027, and…
Ls.160-162: Revise this sentence to eliminate rewordiness.
L.187: Delete “To test for pectinolytic activity,”.
L.192: Delete “To assess the use of nitrates as terminal electron acceptors,”.
L.196: Delete “Solubilization: To determine phosphate solubilizing capacity,”.
L.215: Change “ml/min” by “mL/min”.
L.267-268: Delete this sentence.
L.283: For probability, letter P should be in italic.
Fig.1: But…any statistical analysis? Explain.
Table 1: Again, What statistical analysis did you use to interpret these results?
L.314: Spodoptera frugiperda should be in italic.
Fig.2: There not statistical analysis.
Ls.322-323: This information should be in material and methods section.
L.335: This information should be in material and methods section or delete this sentence.
Table 2: There is a mistake here. For Bacillus tequilensis, verify the RT (or MF) for 1-Nonadecene and Hexadecanoic acid.
Table 3: Also, provide the degree freedom for each ANOVA.
Table 4: Again, provide the degree freedom for each ANOVA.
Table 5: Again, provide the degree freedom for each ANOVA.
L.390: The FAW managing…
Ls.413-414: Change “toxicity” by “lethality”.
Ls.433 and 437, etc.: Overuse of the word “significantly or significant” is not nice. Delete in all manuscript.
Ls.453-458: Any conclusion? You have several results obtained in your investigation but you do not conclude anything. Please rewrite this paragraph.
Author Response
Reviewer 2
These are my main comments on the manuscript (Biology-3275353) entitled “Isolation and Characterization of Bacillus Strains from Egyptian Mangroves: Exploring Their Endophytic Potential in Maize for Biological Control of Spodoptera frugiperda”. This work investigates the colonization potential of indigenous marine Bacillus strains as endophytes in maize plants and insecticidal activity against S. frugiperda. However, details about introduction, materials and methods, and results section are needed. Following substantial revisions should be incorporated in the manuscript prior to acceptance.
L.17: …damaging maize crops and…
Reply: Done
L.19: …of bacteria found in…
Reply: Done
L.22: …particularly Bacillus…
Reply: Done
L.23: …R39 caused significant mortality in laboratory…
Reply: Done
L.44-45: Keywords should be in alphabetic order. Also, keywords serve to widen the opportunity to be retrieved from a database. To put words that already are into title and abstracts makes KW not useful. Please choose terms that are neither in the title nor in abstract.
Reply: Noted with thanks
L.48-50: Summarize this sentence.
Reply: Done
L.51: … historically been a problem in the…
Reply: Done
Ls.53-54: …China and ? Incomplete sentence, revise.
Reply: Corrected
L.54: Sentence starting “The FAW…”.
Reply: Done
L.57: Change “produce” by “products”.
Reply: Done
Ls.57 and 62: Please, abbreviated for FAW. Correct in all manuscript.
Reply: Noted
L.63: Delete “the”.
Reply: deleted
L.73: The word “Bt” should be in italic.
Reply: done
L.88: Again, the word “Bacillus” should be in italic.
Reply: done
L.101: Also, a hypothesis for this study is needed.
Reply: Done
L.131: …ATCC 43300, 3…
Reply: Done extra ; removed
L.132: …ATCC 9027, and…
Reply: Done extra , removed
Ls.160-162: Revise this sentence to eliminate rewordiness.
Reply: Done
L.187: Delete “To test for pectinolytic activity,”.
Reply: deleted
L.192: Delete “To assess the use of nitrates as terminal electron acceptors,”.
Reply: deleted
L.196: Delete “Solubilization: To determine phosphate solubilizing capacity,”.
Reply: deleted
L.215: Change “ml/min” by “mL/min”.
Reply: Changed
L.267-268: Delete this sentence.
Reply: deleted
L.283: For probability, letter P should be in italic.
Reply: Done
Fig.1: But…any statistical analysis? Explain.
Reply: We haven’t done any statistical analysis during the isolation process as we don’t have sample replicates, and all isolates were chosen and purified for further studies.
Table 1: Again, What statistical analysis did you use to interpret these results?
Reply: In order to interpret the data, out of 200 strains, only 33 strains show antagonistic effects against pathogenic bacteria. Statistical analysis was done on all 33 isolates. By using ANOVA, there are no variations in the size of the inhibition zone within the same isolates and the same pathogen (the three replicates have the nearly same measures), while some variations were observed within the same isolate and different pathogens. Regardless of the significance between the same isolate against different pathogens, all 33 isolates were chosen to study their insecticidal activity. As our main objective is to select all strains, that have antagonistic effects even with different inhibition zone diameters.
L.314: Spodoptera frugiperda should be in italic.
Reply: Done
Fig.2: There not statistical analysis.
Reply: Done
Ls.322-323: This information should be in material and methods section.
Reply: Deleted
L.335: This information should be in material and methods section or delete this sentence.
Reply: Deleted
Table 2: There is a mistake here. For Bacillus tequilensis, verify the RT (or MF) for 1-Nonadecene and Hexadecanoic acid.
Reply: Corrected
Table 3: Also, provide the degree freedom for each ANOVA.
Reply: Done
Table 4: Again, provide the degree freedom for each ANOVA.
Reply: Done
Table 5: Again, provide the degree freedom for each ANOVA.
Reply: Done
L.390: The FAW managing…
Reply: Done
Ls.413-414: Change “toxicity” by “lethality”.
Reply: Changed
Ls.433 and 437, etc.: Overuse of the word “significantly or significant” is not nice. Delete in all manuscript.
Reply: Noted
Ls.453-458: Any conclusion? You have several results obtained in your investigation but you do not conclude anything. Please rewrite this paragraph.
Reply: Noted
Reviewer 3 Report
Comments and Suggestions for Authors
This study possesses a significant amount of work, and is interesting because of the positive effect of the presence of Bacillus sp. as an endophyte in maize plants with insecticidal activity against S. frugiperda. However, the reviewer have some questions about the manuscript and could change some statements addressed. So, below, you will find my doubts and other suggestions:
131-132 Please delete unnecessary punctuation.
136-139 Where is the source of the specific method? Please add relevant references.
183-186 Please add relevant references.
300, 303, 314 Why was there no significant analysis in Fig. 1, Fig. 2 and Tab. 1? Please add statistical analysis.
306-313, 335-344 363-381 … It is recommended that the results section be concise and do not list too much data.
382, 385 I suggested that the data in the table be reflected in the form of a graph.
389 The discussion section seems too simplistic,which should highlight the comparison between the author's work and existing works, demonstrating the innovation of the work.
There are many writing format errors. For example, the emergence of microorganisms does not require the full Latin name, but only the first letter of the genus name and the species name. Please review the entire manuscript.
Comments on the Quality of English LanguageThe writing of scientific papers must be impoved.
Author Response
Reviewer 3
This study possesses a significant amount of work, and is interesting because of the positive effect of the presence of Bacillus sp. as an endophyte in maize plants with insecticidal activity against S. frugiperda. However, the reviewer have some questions about the manuscript and could change some statements addressed. So, below, you will find my doubts and other suggestions:
131-132 Please delete unnecessary punctuation.
Reply: Deleted
136-139 Where is the source of the specific method? Please add relevant references.
Reply: Reference added
183-186 Please add relevant references.
Reply: Reference added
300, 303, 314 Why was there no significant analysis in Fig. 1, Fig. 2 and Tab. 1? Please add statistical analysis.
Reply: For fig 1, We haven’t done any statistical analysis during the isolation process as we don’t have sample replicates, and all isolates were chosen and purified for further studies.
For table 1, In order to interpret the data, out of 200 strains, only 33 strains show antagonistic effects against pathogenic bacteria. Statistical analysis was done on all 33 isolates. By using ANOVA, there are no variations in the size of the inhibition zone within the same isolates and the same pathogen (the three replicates have the nearly same measures), while some variations were observed within the same isolate and different pathogens. Regardless of the significance between the same isolate against different pathogens, all 33 isolates were chosen to study their insecticidal activity. As our main objective is to select all strains that have antagonistic effects even with different inhibition zone diameters.
306-313, 335-344 363-381 … It is recommended that the results section be concise and do not list too much data.
Reply: Done
382, 385 I suggested that the data in the table be reflected in the form of a graph.
Reply: The graph has been designed and attached in the supplemental file
389 The discussion section seems too simplistic, which should highlight the comparison between the author's work and existing works, demonstrating the innovation of the work.
Reply: Improved
There are many writing format errors. For example, the emergence of microorganisms does not require the full Latin name, but only the first letter of the genus name and the species name. Please review the entire manuscript.
Reply: Done
Comments on the Quality of English Language
The writing of scientific papers must be impoved.
Reply: Done
Round 2
Reviewer 2 Report
Comments and Suggestions for Authors
The authors have incorporated all suggestions and reviewer comments into the latest version, now the manuscript seems much clear. There are minor points to be corrected:
Ls.43-44: Keywords should be in alphabetic order.
Ls.49, 55-56, 61, 101, 105, 143, 145, 154, 469-470: Please, abbreviated S. frugiperda by FAW.
Ls.468-469, 470-471, 476: Bacillus should be in italic.
Author Response
Reviewer 2
The authors have incorporated all suggestions and reviewer comments into the latest version, now the manuscript seems much clear.
Thank you for your valuable comments.
There are minor points to be corrected:
Ls.43-44: Keywords should be in alphabetic order.
Reply: Done
Ls.49, 55-56, 61, 101, 105, 143, 145, 154, 469-470: Please, abbreviated S. frugiperda by FAW.
Reply: Done
Ls.468-469, 470-471, 476: Bacillus should be in italic.
Reply: Done
Reviewer 3 Report
Comments and Suggestions for Authors
The authors' manuscript has undergone significant improvement after revisions. I think it can meet the publishing requirements of the journal Biology.
Author Response
Reviewer 3
The authors' manuscript has undergone significant improvement after revisions. I think it can meet the publishing requirements of the journal Biology.
Reply: Thank you for your valuable comment.